# TEMPERA: TEST-TIME PROMPT EDITING VIA REINFORCEMENT LEARNING

**Tianjun Zhang[1]**   **Xuezhi Wang[2]**   **Denny Zhou[2]**   **Dale Schuurmans[2, 3]**    **Joseph E. Gonzalez[1]**
[1] UC Berkeley [2] Google Research, Brain Team [3] University of Alberta
`tianjunz@berkeley.edu`

## ABSTRACT

Careful prompt design is critical to the use of large language models in zero-shot or few-shot learning. As a consequence, there is a growing interest in automated methods to design optimal prompts. In this work, we propose **TE**st-ti**M**e **P**rompt **E**diting using **R**einforcement le**A**rning (TEMPERA). In contrast to prior prompt generation methods, TEMPERA can efficiently leverage prior knowledge, is adaptive to different queries, and provides an interpretable prompt for every query. To achieve this, we design a novel action space that allows flexible editing of the initial prompts covering a comprehensive set of commonly-used components like instructions, few-shot exemplars, and verbalizers. The proposed method achieves significant gains compared with recent SoTA approaches like prompt tuning, AutoPrompt, and RLPrompt, across a variety of tasks, including sentiment analysis, topic classification, natural language inference, and reading comprehension. Our method achieves 5.33x on average improvement in sample efficiency when compared to the traditional fine-tuning methods. Our code is available at `https://github.com/tianjunz/TEMPERA`.

## 1 INTRODUCTION

With the recent advances in pre-training large language models (Brown et al., 2020; Fedus et al., 2021; Raffel et al., 2020; Chowdhery et al., 2022), prompting, or in-context learning provides a data-efficient framework for performing NLU (Li & Liang, 2021; Shin et al., 2020b; Gao et al., 2020b). Such methods achieve impressive zero-shot and few-show performance in many downstream tasks.

However, the prompt often has to be carefully tuned to achieve consistent performance for each task (Lu et al., 2021). For example, prompt tuning aims to optimize a continuous prefix embedding via gradient descent and directly takes generated output from the frozen pre-trained language model (Lester et al., 2021; Liu et al., 2021b;a). On the contrary, discrete prompt optimization focuses on constructing meaningful instructions, in-context exemplars and verbalizers (Brown et al., 2020; Gao et al., 2020b). Prior work often performs black-box optimization or applies RL-based methods for direct generation (Deng et al., 2022; Sun et al., 2022; Prasad et al., 2022). Recent works in the prompt tuning field have shown that, performing instance-dependent prompt tuning (Wu et al., 2022; Jiang et al., 2022) can improve the performance of some downstream tasks. The corresponding concept in the discrete prompt optimization domain is intriguing since it allows users to provide different instructions for different inputs and task. Unlike prompt tuning, such instructions can be more human interpretable. However, finding such query-dependent prompts is often overlooked and is not feasible given the inefficiency of black-box optimization.

In this paper, we investigate the importance of providing query-dependent discrete prompts and demonstrate how this can be achieved via efficient search. To this end, we propose the concept of *test-time editing* through reinforcement learning (RL) that allows the agent to perform different editing techniques at *test time* to construct query-dependent prompts efficiently.

We formulate discrete prompt optimization as an RL problem by sequentially editing an initial prompt, which only requires high-level guidance on which part to edit and what tools to use. Different from prior work, this formulation strikes a good balance between human prior knowledge, flexibility, feasibility and interpretability. The method allows easy incorporation of human knowledge since one can provide a manually chosen initial prompt and allow RL to perform editing on

it. It also achieves a balance between search flexibility and feasibility because by enabling different editing techniques, the prompt can be transformed to very different forms but the search space is more feasible compared to direct generation. The final prompt is also more interpretable since the editing tools we adopted usually do not change the semantic meaning of the sentence.

To summarize, we propose to construct query-dependent prompts through test-time editing and formulate this as an RL problem. We carefully design the action space, enabling the agent to flexibly edit the instructions, in-context exemplars and verbalizers. To better train the RL agent, we propose using the score difference between consecutive prompts before and after editing as rewards and developing a set of techniques that help improve the final performance (e.g., reward normalization). We also adopt an attention-based policy architecture to attend over possible candidates or design choices, and show this can be effective for RL training.

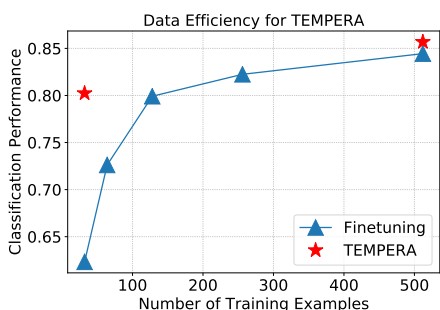

Figure 1: **Data Efficiency for TEMPERA:** We comopare the data efficiency of TEMPERA and standard fine-tuning in a few-shot setting. Results are averaged across four tasks: SST2, AG News, RTE and MR. It shows that our method achieves comparable performance using 4x fewer examples.

Following the standard few-shot text classification setting, we benchmark our algorithm extensively on multiple tasks (including those from GLUE (Wang et al., 2018) and SuperGLUE (Wang et al., 2019)). We show that TEMPERA can achieve SoTA performance (e.g., $1.8\%$ better in SST-2 and $3.9\%$ better in CR) compared to few-shot finetuning, prompt tuning and discrete prompt optimization. We also show that TEMPERA is on 4x more data efficient (over the average of 4 tasks SST2, MR, AG News and RTE) compared with traditional finetuning methods (Figure 1). In addition, we perform extensive ablations on different aspects of the proposed algorithm. We demonstrate that TEMPERA is robust to the prompt pool size and the number of few-shot exemplars.

## 2 RELATED WORK

**Prompting in language models and sensitivity to prompts.** Recent research has shown that as language models scale up, new capabilities could be unlocked such as in-context learning (Brown et al., 2020), where the language model is prompted with a few in-context demonstrations and learns to perform a certain task in a sample-efficient way. However, several works have studied the in-context learning ability more closely and found that the task performance can be highly sensitive to how the in-context prompt is written. For example, Lu et al. (2022) found that the prompt order can have a large effect on the final task performance; Zhao et al. (2021) show that the choice of prompt format, training examples, and prompt order can cause the performance to vary quite significantly.

**Automatic prompt generation and search.** To address such sensitivity in language models, multiple approaches have been proposed for better prompt generation. In the continuous space, Lester et al. (2021) propose prompt-tuning to add tunable tokens for each task during the fine-tuning stage to improve task performance. Zhong et al. (2021) propose OptiPrompt that optimizes the prompts in the input embedding space directly for factual probing. More recently, Wu et al. (2022) found performing instance-independent prompt-tuning can further boost the performance. In the discrete space, Gao et al. (2020a) propose prompt-based fine-tuning and utilize pre-trained models to automatically generate prompt templates. Schick & Schütze (2021) and Schick et al. (2020) use a small amount of training data to automatically identify the best label words to use for few-shot classification. Shin et al. (2020a) propose AutoPrompt to perform gradient-guided search to find the best tokens in the prompt, although the best prompts found are usually not interpretable by humans. Jiang et al. (2020) propose mining-based and paraphrasing-based methods to generate meaningful and diverse prompts for factual knowledge probing. Related to our work, Deng et al. (2022) propose an RL-based framework to directly generate better prompts via black-box optimization. Different from existing work, our approach frames the problem as test-time prompt editing with an RL-based framework to perform efficient search in the editing space.

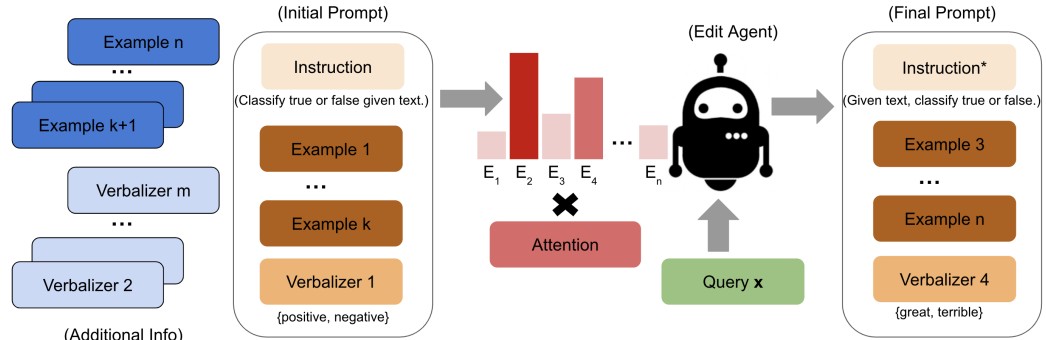

Figure 2: **Test-Time Editing via RL:** The RL agent is trained to optimize the performance of a downstream task. At test-time, given a query, the agent adopts an attention-based policy to edit the instructions, in-context exemplars and verbalizers for $T$ rounds.

**Efficient training exemplar retrieval as prompts.** In addition, existing work has shown the choice of the exemplars can also be critical to the final performance. For example, Liu et al. (2022) propose to retrieve exemplars from a training pool that are semantically similar to a test example, and show it can significantly boost the performance. Rubin et al. (2022) trained a dense retriever to efficiently retrieve good training examples as prompts during test time. In this work, we show that an attention-based exemplar selection process over the embedding space can effectively choose performant training examples within our RL framework.

## 3 TEST-TIME PROMPT EDITING

We formulate the task of test-time editing in this section. We give some background on the few-shot text classification and how to use prompts for downstream NLP tasks. Then we formalize a new setting called *test-time editing* where users are allowed to perform editing over a given prompt, depending on the given input and task during test time.

### 3.1 BACKGROUND

**Few-Shot Text Classification.** Following the standard few-shot language model classification setting (Brown et al., 2020), we assume that we are given a pretrained language model $\mathcal{L}$ and wish to perform classification on dataset $\mathcal{D}$ with label space $\mathcal{Y}$. Assume we are given $K$ samples per class from the training set, the new few-shot training set is given as $\mathcal{D}_{\text{train}} = \{x_i, y_i\}_{i=1}^{K \times |\mathcal{Y}|}$. In addition, there is a hold-out test dataset $\mathcal{D}_{\text{test}}$ that we use for evaluation on downstream NLP tasks.

**Optimizing Discrete Prompts.** Prompt-based few-shot learning considers the following problem: given a piece of text $\mathbf{p}$ as a prompt, we use the generative distribution of the language model $P_{\mathcal{L}}(y|\mathbf{p}, \mathbf{x})$ to perform various NLP tasks without fine-tuning the model. In particular, for a given objective $R$, we propose to perform the desired optimization over the prompt by finding an optimal $\mathbf{p}^* = \arg\min_{\mathbf{p} \in \mathcal{V}} R(P_{\mathcal{L}}(y|\mathbf{p}, \mathbf{x}))$. In this paper, we focus on restricting the prompt $\mathbf{p}$ as a piece of text instead of letting $\mathbf{p}$ to be any vector in the latent space. This not only provides more interpretability of the prompt, but also allows us to use existing natural language tools (e.g., NLTK (Bird et al., 2009)) to perform a discrete search for constructing better prompts.

**Different Forms of Discrete Prompts.** We consider three popular forms of discrete prompts: (1) Instructions, which provide a segment of text describing how the task is performed, usually put at the beginning. (2) In-Context Demonstrations $\{e_0, e_1, ..., e_k\}$, which selects several examples and their corresponding labels, usually placed before the query. (3) Verbalization, which aims to design how the task is asked and which keywords to select as labels. See Figure 2 for an example of different transformations that we perform when editing in our RL-based framework.

### 3.2 TEST-TIME EDITING

---

**Algorithm 1** Test-Time Prompt Editing with TEMPERA

---

1: **Input:** Language Model $\mathcal{L}$, Initial Prompt $p_0$, Training set $\mathcal{D}_{\text{train}}$, Evaluation set $\mathcal{D}_{\text{eval}}$, Iteration $N$, Fix rounds $T$
2: Initialize $\pi_\theta(\cdot \mid s)$ to be uniform;
3: **for** episode $n = 1, \cdots, N$ **do**
4:     Random sample batch $\mathcal{B} \sim \mathcal{D}_{\text{train}}$, Set $p_0$
5:     **for** step $t = 1, \cdots, T$ **do**
6:         Get $s_t = \mathcal{L}(\mathcal{B}, p_t)$
7:         Run editing policy $a_t = \pi_\theta(s_t)$, Get new prompt $p_{t+1}$
8:         Get new state $s_{t+1} = \mathcal{L}(\mathcal{B}, p_{t+1})$
9:         Add transition $(s_t, a_t, s_{t+1})$ to replay buffer
10:    **end for**
11:    Update policy parameter $\theta$ of $\pi_\theta$ with the PPO loss
12: **end for**
13: **Evaluate** policy $\pi_\theta$ on evaluation dataset $\mathcal{D}_{\text{eval}}$

---

Prior works have often attempted to identify a query-agnostic prompt (Deng et al., 2022; Sun et al., 2022) or attempted to directly generate a query-dependent prompt via hyper-networks learning (He et al., 2022). However, query-agnostic prompting fails to incorporate any query-related information into the prompts and directly generating prompts for each individual query is challenging (due to its difficulty to incorporate human prior knowledge or feedback). In addition, by permuting the order of in-context exemplars $\{e_0, e_1, ..., e_k\}$ (Lu et al., 2022) or searching for the $k$ nearest neighbors of the current test instance (Liu et al., 2022) as in-context exemplars yields better performance. These reveal the importance of constructing query-dependent prompts.

Unlike prior methods, we perform prompt editing at test-time. The procedure works as follows: at test time, one is given an initial prompt $\mathbf{p_0}$. We want to learn a function $f$ that takes the initial prompt $\mathbf{p_0}$, query $\mathbf{x}$ and a pool of examples/verbalizers $\mathbf{p'}$, and outputs a final prompt: $\mathbf{p_f} = f(\mathbf{p_0}, \mathbf{x}, \mathbf{p'})$. The overall framework of our algorithm is shown in Fig. 2. We allow $f$ to make edits (e.g., editing verbalizers and/or swapping examples) over the original prompt to make it more suitable for the downstream task and query $\mathbf{x}$. Since the editing function $f$ can depend on the query $\mathbf{x}$, we call it the test-time editing function. Note that we train the function $f$ in a fixed training dataset and directly deploy it at test time without any addition training. This is different from the test-time optimization since we don't have access to the ground truth label or a surrogate objective. Plese see Algorithm.1 for details.

## 4 TEST-TIME EDITING VIA REINFORCEMENT LEARNING

In order to learn the test-time editing function $f$, we present a novel RL-based framework that naturally maps the editing process to an MDP. We will present our framework and discuss how we design the state space, action space and reward in this section.

**Reinforcement Learning Formulation.** We formulate test-time editing as a Markov Decision Process (MDP). Given an initial state, $\mathbf{s} = (\mathbf{p_0}, \mathbf{x})$, consisting of an initial prompt and a query, at each time step $t$, the RL agent selects one of the editing methods from the action space $A$. We can then define the transition function $\mathcal{T} : S \times A \to S$ to be the state of prompt before and after editing $(\mathbf{p_t}, \mathbf{x}) \times \mathbf{a_t} \to (\mathbf{p_{t+1}}, \mathbf{x})$. That is, the transition dynamics are deterministic given the editing action. We can either define a fixed horizon $H$ or design a termination function to stop editing and get the final prompt. The goal is to maximize the expected reward $R = \mathbb{E}[\sum_{k=0}^{T} \gamma^k r_k]$ where $r_t$ is the reward and $\gamma$ is the discount factor. We introduce in detail each component of the state representation, action space and rewards in the following subsections.

**State Representation.** The RL framework is general and flexible about the representation of states. The only requirement is that such representation contains text information. Instead of directly using the raw text representation, we use the last hidden states of the pretrained language model $\mathbf{s_t} = \mathcal{L}(\mathbf{p_t}, \mathbf{x})$ as the state representation and feed it into the policy network.

Table 1: Effect of different editing techniques. For instruction, we tokenize it into phrases and perform swapping, addition or deletion. We also allow swapping in-context exemplars or changing different verbalizers.

| | | Before Editing | After Editing |
|---|---|---|---|
| Instruction | Swap | "Given text, classify whether it is good or bad." | "Classify whether it is good or bad, given text." |
| | Add | "Given text, classify whether it is good or bad." | "Given text, given text, Classify whether it is good or bad." |
| | Delete | "Given text, classify whether it is good or bad." | "Classify whether it is good or bad." |
| Example | Permute | {Example 1, Example 2, ..., Example $k$ } | {Example $k$, Example 3, ..., Example 1 } |
| | Swap | {Example 1, Example 2, ..., Example $k$ } | {Example $k+1$, Example $n$, ..., Example 1 } |
| Verbalizer | Change | { "positive", "negative"} | {"great", "terrible"} |

**Action Space Design.** We include most of the editing actions in our action space. At each stage, the RL agent can choose the editing objects from instruction, in-context exemplars or verbalizer. For editing the instruction, we provide the initial instruction from natural instructions (Wang et al., 2022). Then we tokenize the instruction into phrase level using NLTK (Bird et al., 2009) and perform swapping, deletion or addition of different phrases. Suppose we have $l$ phrases, the action space size will become $(l \times (l-1))/2 + 2l$.

For the in-context exemplars, we keep an example pool of $N$, initialize our prompt by randomly choose $n$ of them as the initial prompt. We then allow the agent to directly perform swapping one example from the current prompt with either another one from the current prompt or from the pool of examples that are not currently used. This results in an action space for the RL agent of $n \times N - (n \times (n-1))/2$ since we do not allow swapping with the same example.

For the verbalizer, we allow the RL agent to freely choose which verbalizer to use for each in-context example from PromptSource (Bach et al., 2022). We also will enable the agent to freely choose which verbalizer to use for each query $\mathbf{x}$. Interestingly we found that this helps boost the performance of our algorithm. We provide some examples of the editing process in Tab. 1.

**Reward Design.** We adopt the step reward proposed in RLPrompt (Deng et al., 2022). For each query $\mathbf{x}$, we get the log probability of the output label from the language model $\log P_{\mathcal{L}}(\hat{y}|\mathbf{x}, \mathbf{p_t})$ given the proposed prompt $\mathbf{p_t}$ with the correct label $c$, and we define the score difference $s(c)$ as:

$$s(c, \mathbf{x}, \mathbf{p_t}) = \lambda_1 \log P_{\mathcal{L}}(\hat{y}_c|\mathbf{x}, \mathbf{p_t}) - \lambda_2 \arg\max_{c' \neq c} \log P_{\mathcal{L}}(\hat{y}_{c'}|\mathbf{x}, \mathbf{p_t}) \tag{1}$$

where we have introduceed the two balancing hyperparameters $\lambda_1 > 0$ and $\lambda_2 > 0$ for the positive and negative terms respectively. Intuitively, this score gives a negative reward when the prediction is not correct and a positive reward otherwise. The goal is to optimize the score for the final prompt.

However, RL aims to optimize the accumulated reward during the MDP process while prompt design only cares about the performance of the final prompt. Thus, we propose to use the score difference between successive edits as the immediate reward:

$$r_t = s(c, \mathbf{x}, \mathbf{p_t}) - s(c, \mathbf{x}, \mathbf{p_{t-1}}) \tag{2}$$

Ignoring the discounting factor $\gamma$, this makes the accumulated reward from time 0 to $T$ correspond to the score difference between the final and the initial prompt $s(c, \mathbf{x}, \mathbf{p_T}) - s(c, \mathbf{x}, \mathbf{p_0})$. Now the objective of RL is to maximize the score difference.

**Attention-Based Policy Architecture.** We adopt an attention-based policy architecture for the reinforcement learning agent. We put attention over a graph of possible candidates and let the agent choose which editing technique to perform. We find that the attention-based architecture helps the agent to emphasize the important examples (e.g., examples that are more semantically similar to the test instance).

We use the PPO (Schulman et al., 2017) algorithm in our experiments. The detailed hyperparameter used can be found in Appendix. A. We list here a couple of very important techniques we used in our experiments. We found these techniques are crucial to the success of our RL-based framework.

**Observation Normalization**: Since we take the last hidden states of the language model as observation, it might have very small variances between different samples. We keep a running mean

and standard deviation for the observation and normalize it before feeding it to the policy and value network. This is commonly used in RL and we found this boosts the performance of our method.

**Reward Normalization**: For different training samples, performing editing over prompts may result in significantly different reward scales. For some of the samples, different prompts might have very marginal effects on the final prediction, either due to the fact that the model is already confident about the prediction since it is too easy, or the task sample is too hard to predict and the model is confused regardless of what prompt it is fed. On the other hand, for other training samples, editing prompts might bring a huge difference in terms of the accuracy. Thus, we perform sample-wise reward normalization to ensure that the reward scale between samples is relatively consistent.

**Conditioning Policy on Action History**: Directly taking the observation from the language model can be inefficient since the policy has no clue about how it has reached the current state. This will bring a loop that the policy will edit prompts $\mathbf{p}_A \to \mathbf{p}_B$ and then $\mathbf{p}_B \to \mathbf{p}_A$. To mitigate this effect, we build a policy that not only takes in the current hidden state, but also conditioned on the action history on how it gets to the current state. Thus, we break the loop between two prompts by considering how each state is reached.

## 5  EXPERIMENTS

Our experiments first reveal the effectiveness of TEMPERA in the few-shot setting. We compare TEMPERA with prior baselines like Finetuning (Devlin et al., 2019), Soft Prompt Tuning (Lester et al., 2021), Black-Box Tuning (Sun et al., 2022), RLPrompt (Deng et al., 2022) and other manually tuned prompt methods. On various tasks from GLUE (Wang et al., 2018) and SuperGLUE (Wang et al., 2019), our method achieves impressive performance comparing to prior baselines. This shows that only using a small amount of training examples is sufficient for RL and TEMPERA is sample efficient. We also illustrate the data efficiency of our method compared to finetuning, showing that TEMPERA can achieve same performance with 5.33x less data.

In addition to the performance gains, we aim to understand our method from different aspects. In Sec. 5.2, we study how much test-time editing helps compared to query-agnostic prompts. Our experiments demonstrate the importance of test-time editing and the necessity of query-dependent prompts. In Sec. 5.4, we show that how different editing techniques (e.g, instruction, in-context demonstration and verbalization) affect the final performance of the downstream task. We also ablate the number of in-context demonstrations used and the size of the example pool in Sec. 5.6 and Sec. 5.7. Finally, we show some example prompts after editing to illustrate the editing policy.

**Tasks.**    We conduct our experiments from different categories including single-sentence tasks (e.g., sentiment analysis including SST-2, Yelp reviews, MR, CR, topic classification including AG News). For one-sentence tasks, the goal is to make a prediction based on the sentence. We also include tasks from different types like NLI (e.g., SST-2) and multiple choices (e.g., AG News). Most of the tasks are from the standard GLUE (Wang et al., 2018).

**Task Settings.**    To ensure a fair comparison, we follow the same setting from LM-BFF (Gao et al., 2020b) and RLPrompt (Deng et al., 2022), we test TEMPERA on few-shot text classification tasks. The setting is devised as follows: We randomly sample 16 training samples per class from the training dataset of each task and use them as the few-shot dataset. This will result in a total of $16 \times |\mathcal{Y}|$ training samples (please refer to Appendix. E for the number of classes in each task). We also randomly sample 16 samples per class as the validation dataset. For reporting the final performance, we use the standard test set and the detailed information can be found at Appendix E. In addition to the common setup, we also randomly select $n$ examples from the training dataset as the in-context exemplar pool. We average our runs for 4 random seeds and report the average performance and corresponding standard deviation. For the language model, we use $\mathcal{L} = \text{RoBERTa-large}$ (Liu et al., 2019). For the details of these settings and tasks, please refer to Appendix. E. The initial instruction is taken from the Natural Instructions (Mishra et al., 2021). The initial in context demonstrations are randomly sampled from a fixed example pool of size 16 and the example pool is also randomly sampled from the training dataset, different from the few-shot dataset that used for training the RL policy.

**Baselines.** We compare TEMPERA with several SoTA prompt tuning and discrete prompt optimization baselines (including finetuning).

- **Finetuning:** it finetunes the entire language model with a classification head using the few-shot dataset.
- **Manual Prompt:** we take the handcrafted prompt from (Bach et al., 2022).
- **Black-Box Tuning:** it is a mixture of discrete and soft prompt. The soft part is trained using gradient descent and the discrete part is optimized using gradient-free tuner.
- **AutoPrompt:** it adds the discrete trigger token and updates the prompts by iterative gradient search.
- **In-Context Demonstration:** it randomly selects one training example and concatenates them with the input query.
- **Instructions:** Following Natural Instructions (Wang et al., 2022), prompts are manually created instruction for each task. Each prompt is concatenated with inputs. Details are in Appendix. D.
- **GrIPS:** it performs phrase level editing on the instructions and selects the best one.
- **RLPrompt:** it generates discrete prompts using RL framework.

## 5.1 Few-Shot Text Classification

Following the settings in existing work, we evaluate our model on some few-shot text classification tasks. In Tab. 2, We compare our method with various baselines including RLPrompt. We can see that on most tasks we tested, TEMPERA outperforms previous baselines by a large margin. For example, we have a 1.8% absolute gain on the SST-2 task (over RLPrompt), 3.9% gain on the CR task and the performance is almost comparable to finetuning the language model on the AG News task. We also see that our method results in a much smaller variance between runs than Soft Prompt Tuning and AutoPrompt, indicating that it is more stable across different few-shot datasets. Comparing to search-based methods (e.g., Black-Box Tuning or GrIPS), our method avoids the expensive run-time search if one wants to perform test-time editing using one of the black-box optimization methods with a surrogate reward. Note since the original Black-Box Tuning or GrIPS paper didn't perform query-dependent search, this is our conjecture. Thus, out method achieves both test-time efficiency and good performances on downstream tasks.

Table 2: Few-shot classification results. We compare against different baselines in this setting. Results show that TEMPERA surpasses various baselines including finetuning, prompt tuning and discrete prompt search. The standard deviations are shown in brackets.

| | | SST-2 | Yelp P. | MR | CR | AG News |
|---|---|---|---|---|---|---|
| Finetuning | Finetuning (few-shot) | 80.6 (3.9) | 88.7 (4.7) | 67.4 (9.7) | 73.3 (7.5) | 84.9 (3.6) |
| Continuous Prompt | Soft Prompt Tuning | 73.8 (10.9) | 88.6 (2.1) | 74.1 (14.6) | 75.9 (11.8) | 82.6 (0.9) |
| | Black-Box Tuning | 89.1 (0.9) | 93.2 (0.5) | 86.6 (1.3) | 87.4 (1.0) | 83.5 (0.9) |
| | AutoPrompt | 75.0 (7.6) | 79.8 (8.3) | 62.0 (0.8) | 57.5 (5.8) | 65.7 (1.9) |
| Discrete Prompt | Manual Prompt | 82.8 | 83.0 | 80.9 | 79.6 | 76.9 |
| | In-Context Demo. | 85.9 (0.7) | 89.6 (0.4) | 80.6 (1.4) | 85.5 (1.5) | 74.9 (0.8) |
| | Instructions | 89.0 | 84.4 | 85.2 | 80.8 | 54.8 |
| | GrIPS | 87.1 (1.5) | 88.2 (0.1) | 86.1 (0.3) | 80.0 (2.5) | 65.4 (9.8) |
| | RLPrompt | 90.1 (1.8) | **93.9 (1.8)** | 86.7 (2.4) | 87.2 (1.7) | 77.2 (2.0) |
| Discrete Prompt | TEMPERA (ours) | **91.9** (2.0) | 92.6 (1.7) | **88.0** (1.1) | **91.1** (1.6) | **85.5** (1.5) |

## 5.2 Importance of Test-time Prompt Editing

To illustrate the importance of test-time prompt editing, we compare our method with various baselines that do not perform test-time editing. In addition, we also construct another baseline where we

Figure 3: **Data Efficiency for TEMPERA:** We compare data efficiency between TEMPERA and few-shot finetuning. Results show that we can achieve a good performance with significantly less data (varying from 4x to 8x).

create a RL based method where the policy is not dependent on the input query $x$, denoted as "TEMPERA (No TTE)". Results in Tab. 3 show that TEMPERA even without test-time editing can find better query-agnostic prompts comparing to manually construct prompts, in-context demonstration and GrIPS. However, adding test-time editing can further improve the performance when the task is harder: we got $0.8\%$ improvement on MR task and $3.0\%$ improvement at AG News task. On SST-2, the effect of test-time editing is not significant as we suspect that the task is too easy. We found on harder tasks like AG News, the gain of test-time editing is huge.

Table 3: We compare our method against different methods which do not perform test-time editing. Results show that test-time editing is mostly helpful in harder tasks like AG News.

|                  | SST-2 | MR   | AG News |
|------------------|-------|------|---------|
| Manual Prompt    | 82.8  | 80.9 | 76.9    |
| In-Context Demo. | 85.9  | 80.6 | 74.9    |
| Instructions     | 89.0  | 85.2 | 54.8    |
| GrIPS            | 87.1  | 87.1 | 65.4    |
| TEMPERA (No TTE) | **92.0** | 87.4 | 81.3    |
| TEMPERA          | 91.9  | **88.2** | **84.3** |

Table 4: Ablation on different editing techniques. Results show that adding verbalizer-edits helps all the tasks (especially MR and AG News). Adding instruction-edits marginally helps the performance in SST-2 and MR.

|                              | SST-2 | MR   | AG News |
|------------------------------|-------|------|---------|
| TEMPERA (No Inst & Verb)     | 91.2  | 87.2 | 82.2    |
| TEMPERA (No Inst)            | 91.9  | 88.2 | 84.3    |
| TEMPERA                      | **92.4** | **88.4** | **85.5** |

## 5.3 DATA EFFICIENCY FOR TEMPERA

To illustrate the data efficiency of our method, we compare the performance of TEMPERA with some few-shot standard finetuning results in Fig. 3. We see that in SST-2, we achieve similar performance using almost 8x fewer training data. In tasks like Yelp, the gain is about 4x. We see that with fewer examples, TEMPERA strictly dominates fine-tuning methods. This is critical when applying TEMPERA in the real-world application since labeled data is expensive to get.

## 5.4 QUALITATIVE ANALYSIS OF THE EDITS

We also visualize our policy by taking a few examples from the final prompts after editing in Tab. 5. We see that our method mostly does example selection, verbalizer swapping and phrase-level instruction editing. Our editing techniques are flexible and the final prompt may take different combinations for each query. In addition, the resulting final prompt is still interpretable by human, showing that our method achieves flexibility and interpretability at the same time. Note that in the examples provided in Tab. 1, our policy choose to modify the example selection and verbalization.

## 5.5 ABLATION: DIFFERENT EDITING TECHNIQUES

We ablate on the different editing techniques and study how adding or removing them can affect the performance. The results are shown in Tab. 4. We can see that adding each component (e.g., verbalizer, instruction) is helpful in terms of the final performance. We also find that verbalizer is especially helpful in some tasks like AG News, resulting in a 1.2% difference in the final performance. This indicates that adding more flexibility to some extent can help the performance.

Table 5: Qualitative results on the effect of the learned policy. We see that our method both enables the flexibility of various edits and interpretability of the final results. On the contrary, many prior methods produce non-readable prompts. Red text is prior to editing and blue text are the changes.

| SST-2 | Before Edit | "In this task, you are given sentences from movie reviews. The task is to classify a sentence as "positive" if the sentiment of the sentence is positive or as "negative" if the sentiment of the sentence is negative. Review: of saucy. Sentiment: positive. Review: cold movie. Sentiment: negative. Review: heroes. Sentiment: <mask>." |
| | After Edit *(better verbalizer)* | "In this task, you are given sentences from movie reviews. The task is to classify a sentence as "great" if the sentiment of the sentence is positive or as "terrible" if the sentiment of the sentence is negative. Review: of saucy. Sentiment: great. Review: cold movie. Sentiment: terrible. Review: heroes. Sentiment: <mask>." |
| AG News | Before Edit | "Classify the news articles into the categories of World, Sports, Business, and Technology. Article: What's in a Name? Well, Matt Is Sexier Than Paul (Reuters) Reuters - As Shakespeare said, a rose by any other name would smell as sweet. Right? Answer: Technology. Article: Wall St. Bears Claw Back Into the Black (Reuters) Reuters - Short-sellers, Wall Street's dwindling band of ultra-cynics, are seeing green again. Answer: <mask>." |
| | After Edit *(better exemplar selection)* | "Classify the news articles into the categories of World, Sports, Business, and Technology. Article: Expansion slows in Japan Economic growth in Japan slows down as the country experiences a drop in domestic and corporate spending. Answer: Business. Article: Wall St. Bears Claw Back Into the Black (Reuters) Reuters - Short-sellers, Wall Street's dwindling band of ultra-cynics, are seeing green again. Answer: <mask>." |

Table 6: Ablation on the number of in-context exemplars. Results show that increasing the number of examples results in a consistent increase of performance except for AG News (which is due to the length limit).

| | SST-2 | MR | AG News |
|---|---|---|---|
| TEMPERA (2 Examples) | 91.6 | 87.9 | 84.0 |
| TEMPERA (4 Examples) | 91.9 | 88.2 | **84.3** |
| TEMPERA (8 Examples) | **92.4** | **88.4** | 82.2 |

Table 7: Ablation on the size of the prompt pool to select from. We see that the performance does not change too much when changing the size of the pool, indicating that the performance is relatively stable.

| | SST-2 | MR | AG News |
|---|---|---|---|
| TEMPERA (Pool Size 8) | 91.6 | 87.9 | 84.1 |
| TEMPERA (Pool Size 16) | 91.9 | 88.2 | 84.3 |
| TEMPERA (Pool Size 32) | **92.2** | **88.4** | **84.7** |

## 5.6 ABLATION: NUMBER OF SHOTS

We also ablate on the number of examples used in the in-context demonstration part of our algorithm. We choose the size of 2, 4 and 8 for the analysis. We see that from Tab. 6, in all the tasks we tested (SST-2, MR and AG News), increasing the number of examples consistently improves the performance. However, the performance improvement is relatively limited. In addition, due to the input length limit constraint by the language model (512 for RoBERTa), longer sequences of input will be truncated. This results in the performance decrease when increasing the number of examples from 4 to 8 for AG News, where the input length is longer than 512.

## 5.7 ABLATION: SIZE OF THE PROMPT POOL

We also ablate on the example size of the prompt pool where we keep the number of examplers of 4. Intuitively, allowing our method to choose in-context demonstrations from a large range of example pool can provide better prompts. From Table. 7, we can see that increasing the example pool size gives the algorithm more flexibility to choose in-context demonstrations, resulting in a slightly better final performance.

## 6 CONCLUSION

In this paper we present TEMPERA, a test-time prompt editing method for large language models via reinforcement learning. We found that perform test-time editing can greatly improve the performance of downstream tasks for a pretrained language model. The proposed method only requires little guidance on high-level search space design and can easily incorporate prior human knowledge. It achieves SoTA performance on multiple benchmarks including those from GLUE. This intersection area of research between NLP and RL can inspire future research on designing better test-time editing algorithms for practical usage.

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

## A  TRAINING DETAIL

We provide the training details here. We use standard PPO algorithm to do online policy optimization with GAE. We provide all the hyperparameters here for a reference. We'll specify our neural network architecture in the following section. Note that we perform additional observation normalization (i.e., keeping a running mean and std) and reward normalization. We also adopt the same number of parallel environment as the few-shot setting (e.g., 32 in our few-shot experiments). We found a large size of parallel environment helps boost the performance.

Table 8: Hyperparameters used for TEMPERA in all the tasks.

|  | Hyperparameter Value |
| --- | --- |
| Steps per training | 8 |
| Time limit | 8 |
| Number Parallel Processes | 256 |
| Learning rate | 0.00005 |
| Entropy Coefficient | 0.005 |
| Value loss Coefficient | 0.5 |
| Mini Batch Size | 32 |
| Gamma | 0.99 |
| GAE Lambda | 0.95 |
| Number of in-context Exemplars | 4 |
| Number of example pool | 16 |
| Positive lambda coefficient ($\lambda_1$) | 2.0 |
| Negative lambda coefficient ($\lambda_2$) | 1.8 |

## B  NETWORK ARCHITECTURE

We follow the GPT (Brown et al., 2020) architecture and use the encoder layer for our policy network. Note that our policy and baseline network shares the same attention-based encoder. The attention is flat over all the possible candidate examples. We use a 3-layer encoder block with 3 heads and 48 latent dimension. We build two different head with 2-layer MLP for each as the policy head and baseline head. We also don't use dropout for the policy learning part. We found this boost up the performance.

## C  ADDITIONAL EXPERIMENTS

We perform additional experiments on some more tasks like RTE, QNLI, SNLI, MNLI and MRPC. Results show that we are consistently better than most of the discrete prompt optimization methods and continuous prompt tuning methods. On several tasks, we are also better than finetuning the entire model.

## D  NATURAL INSTRUCTIONS AND PROMPTSOURCE

We provide all the instructions we used in our experiments from Natural Instructions. Here we just provide a few examples. Please refer to the github for all the instruction they provided. We also provide all the verbalizers we used in our experiments from Promptsource. Here we only provide a few examples. Please also refer to their github for the full verbalization.

## E  DATASET DETAIL

For the Finetuning, we use standard finetuning of the RoBERTa model from huggingface for 100 epochs, a learning rate of 0.0003 and the optimizer of Adam.

Table 9: Few-shot classification results. We compare against different baselines in this setting. Results show that TEMPERA surpasses various baselines including finetuning, prompt tuning and discrete prompt search. The standard deviations are shown in brackets.

| | | RTE | QNLI | SNLI | MNLI | MRPC |
|---|---|---|---|---|---|---|
| Finetuning | Finetuning (few-shot) | 58.6 (3.9) | 60.2 (4.7) | 54.64 (9.7) | 47.8 (7.5) | 77.4 (3.6) |
| Continuous Prompt | Soft Prompt Tuning | 54.7 (10.9) | 49.7 (0.2) | 36.13 (14.6) | 33.2 (0.0) | 51.6 (0.9) |
| | Black-Box Tuning | 52.6 (0.9) | 48.8 (0.6) | 46.58 (1.3) | 42.9 (2.0) | 61.6 (0.9) |
| Discrete Prompt | Manual Prompt | 51.6 | 50.8 | 31.11 | 51.7 | 67.4 |
| | In-Context Demo. | 60.4 (0.7) | 53.8 (0.4) | 47.11 (1.4) | 53.4 (1.5) | 45.8 (0.8) |
| Discrete Prompt | TEMPERA (ours) | 60.3 (2.2) | 57.4 (1.5) | 56.4 (3.2) | 45.2 (2.0) | 74.0 (1.0) |

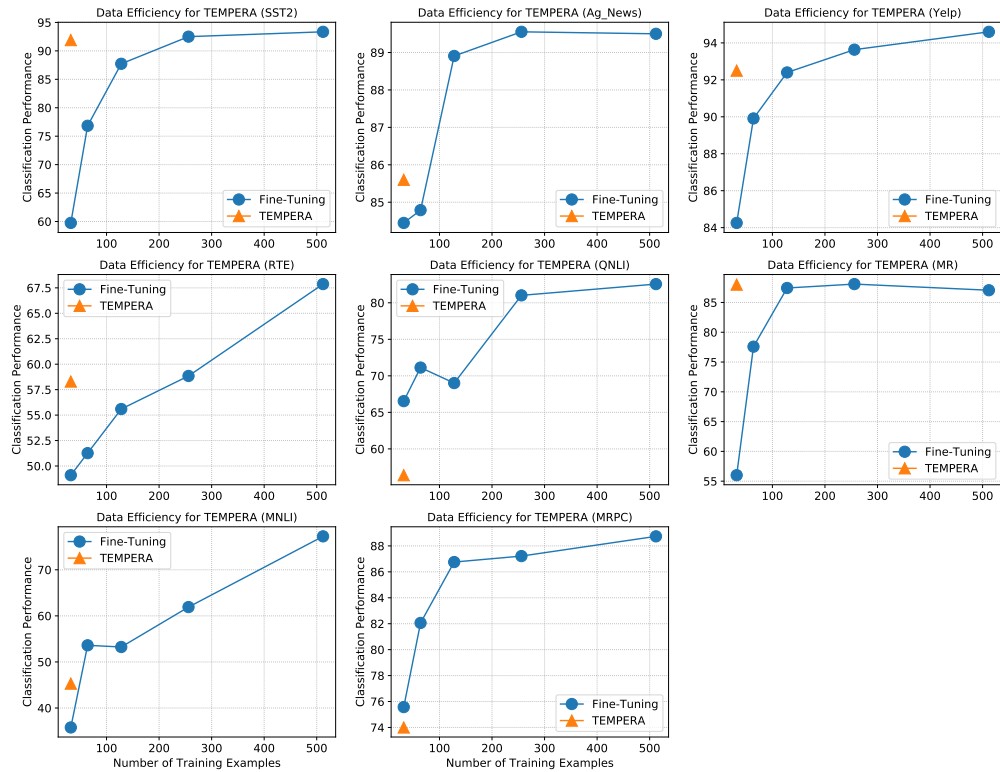

Figure 4: **Data Efficiency for TEMPERA:** We plot all the finetuning performance for 8 tasks we tested. We see that TEMPERA often achieves the better few-shot performance except for MRPC and QNLI.

## F  COMPARISON OF DIFFERENT METHOD

We compare the different property of different prompting methods in this section in order to give a better understanding of different algorithms.

Table 10: Natural instructions used for TEMPERA in all the tasks.

| Task | Natural Instructions |
| --- | --- |
| SST-2 | "In this task, you are given sentences from movie reviews. The task is to classify a sentence as "great" if the sentiment of the sentence is positive or as "terrible" if the sentiment of the sentence is negative." |
| AG News | "Classify the news articles into the categories of World, Sports, Business, and Technology." |
| CR | "In this task, you are given sentences from customer reviews. The task is to classify a sentence as "great" if the sentiment of the sentence is positive or as "terrible" if the sentiment of the sentence is negative." |
| MR | "In this task, you are given sentences from movie reviews. The task is to classify a sentence as "great" if the sentiment of the sentence is positive or as "terrible" if the sentiment of the sentence is negative." |
| Yelp | "In this task, you are given sentences from Yelp reviews. The task is to classify a sentence as "great" if the sentiment of the sentence is positive or as "terrible" if the sentiment of the sentence is negative." |
| RTE | N/A |
| SNLI | "In this task, you're given a pair of sentences, sentence 1 and sentence 2. Your job is to choose whether the two sentences clearly agree (entailment)/disagree (contradiction) with each other, or if this cannot be determined (neutral). Your answer must be in the form of the letters Yes, Maybe, and No respectively." |
| QNLI | "You are given two sentences(Sentence1 and Sentence2). Answer "yes" if these sentences are a paraphrase of one another, otherwise answer "no"." |
| MNLI | "In this task, you're given a pair of sentences, sentence 1 and sentence 2. Your job is to choose whether the two sentences clearly agree (entailment)/disagree (contradiction) with each other, or if this cannot be determined (neutral). Your answer must be in the form of the letters Yes, Maybe, and No respectively." |

Table 11: Verbalizers used for TEMPERA in all the tasks.

| Task | Natural Instructions |
| --- | --- |
| SST-2 | 'Someone just said to me "{{sentence}}". Do you think they are {{"sad"}} or {{"happy"}}? {{ answer_choices[label]}}' |
| AG News | "What label best describes this news article? {{text}} {{answer_choices[label]}}" |
| CR | 'Someone just said to me "{{sentence}}". Do you think they are {{"sad"}} or {{"happy"}}? {{ answer_choices[label]}}' |
| MR | '{{text}} Did the reviewer find this movie {{"good or bad"}}? {{ answer_choices[label] }}' |
| Yelp | '{{ text }} Overall, the experience is {{ answer_choices[label] }}' |
| RTE | 'Does the claim "{{sentence2}}" follow from the fact that "{{sentence1}}"? Please answer either {{"yes"}} or {{"no"}}. {{answer_choices[label]}}' |
| SNLI | 'Suppose {{premise}} Can we infer that "{{hypothesis}}"? Yes, no, or maybe? {{ answer_choices[label] }}' |
| QNLI | '{{sentence}} Does that sentence have all you need to answer the question "{{question}}"? {{answer_choices[label]}}' |
| MNLI | 'Suppose {{premise}} Can we infer that "{{hypothesis}}"? Yes, no, or maybe? {{ answer_choices[label] }} ' |
| MRPC | 'Does the sentence {{sentence1}} paraphrase (that is, mean the same thing as) this sentence? {{sentence2}} {{ answer_choices[label] }}' |

Table 12: Scaling results for TEMPERA in 512 training data per class. Results show that TEMPERA also scales and achieves better results comparing to finetuning.

|  |  | SST2 | MR | AG News | RTE |
|---|---|---|---|---|---|
| Finetuning | Finetuning (few-shot) | 93.4 | 87.0 | **89.5** | 67.9 |
| Discrete Prompt | TEMPERA (ours) | **93.8** | **88.6** | 88.6 | **71.4** |

Table 13: Details for the dataset including the type, size of training, evaluation and test. Note that here all the sizes are few-shot dataset.

| Dataset | Type | $|C|$ | $|\text{Train}| = |\text{Dev}|$ | $|\text{Test}|$ |
|---|---|---|---|---|
| SST2 | Sentiment | 2 | 32 | 1.8k |
| AG News | topic | 4 | 64 | 7.6k |
| CR | Sentiment | 2 | 32 | 2k |
| MR | Sentiment | 2 | 32 | 2k |
| Yelp | Sentiment | 2 | 32 | 38k |
| RTE | NLI | 2 | 32 | 0.3k |
| SNLI | NLI | 3 | 48 | 10k |
| QNLI | NLI | 3 | 48 | 9.8k |
| MNLI | NLI | 3 | 48 | 9.8k |

| | Frozen LM | Gradient-Free | Guided-Optimization | Interpretable | Query-Dependent |
|---|---|---|---|---|---|
| Fine-Tuning | ✘ | ✘ | ✔ | ✔ | ✘ |
| Manual Prompt | ✔ | ✔ | ✘ | ✔ | ✘ |
| Instructions | ✔ | ✔ | ✘ | ✔ | ✘ |
| In-Context Demonstration | ✔ | ✔ | ✘ | ✔ | ✘ |
| Soft Prompt Tuning | ✔ | ✘ | ✔ | ✘ | ✘ |
| Discrete Prompt Search | ✔ | ✔ | ✘ | ✔ | ✘ |
| AutoPrompt | ✔ | | ✔ | ✘ | ✘ |
| RLPrompt | ✔ | ✔ | ✔ | ✘ | ✘ |
| Tempera (ours) | ✔ | ✔ | ✔ | ✔ | ✔ |

Figure 5: **Comparison of Different Prompting Methods:** We compare the different property of different algorithms. We can see that TEMPERA is gradient-free, the resulting prompt is interpretable and query-dependent.

