# OpenReview forum: "TEMPERA: Test-Time Prompt Editing via Reinforcement Learning"
_ICLR.cc/2023/Conference — ICLR 2023 notable top 25%_

### Official Review · Reviewer_Hk2t · 2022-10-22

**Confidence:** 4
**Correctness:** 3
**Technical Novelty And Significance:** 3
**Empirical Novelty And Significance:** 3
**Recommendation:** 8

**Clarity, Quality, Novelty And Reproducibility:**

Clarity:
While the description of related work and where the method is positioned, background, and the discussion of the results is clear, I am missing several details and clarification questions in the method section, which are the main reason for my rating. I would strongly suggest editing the text and figures to reflect those.

Novelty:
While using RL to improve prompts has been used in other works, conditioning the prompt search on the query, and the action space for editing are novel and effective.

Reproducibility:
Clarifying aspects of the method would help in reproducibility, as well as providing the code. On the other hand, the appendix contains details about training hyper-parameters and datasets, which help in reproducibility. Correcting the clarity questions will improve reproducibility.

**Strength And Weaknesses:**

**Strengths:**
- Relevance:
	- Given the wide use of prompting for tasks in LLMs, and the effect it can have in final performance of downstream tasks, exploring methods to design better prompts is a very important problem.
- Writing:
	- I like how the related work is structured, motivating first the need for prompt learning, and later providing an overview of prompt learning methods and methods based on retrieval examples from the training set.
	- Formulation in section 4 makes the method clear to understand and connect to RL frameworks.
	- Table 1 is useful to understand action space
- Method:
	- The policy used to modify prompts is constrained, which may allow for more interpretability and sampling efficiency (Fig 1), while at the same rich in some parts (swap example, verbalizer change), allowing different kinds of operations ito the prompts.

- Experiments:
	- Thorough and valuable ablation studies, comparison with solid baselines and in different tasks.
	- SOTA or comparable results in a wide set of tasks.
	- Very good properties in few-shot settings.
	- Very cool example of exemplar editing in Table 5, I would love to see this together with the other exemplars. In general, I think it would be really interesting to see if there are common properties in the exemplars edited by the policy. Does the model tend to balance the number of exemplars per class, or does it try to select more exemplars close to the query class?


**Weaknesses:**


- Writing Clarity
	- From my understanding of the paper, the prompts are edited at test time using a pre-trained policy, but the policy is not updated at test-time but training time. While this is reasonable because at test time we do not have access to the GT label and therefore cannot compute the reward, the test-time optimization generally makes me think about having a surrogate reward at test time used to update the policy. I would strongly encourage to clarify this more explicitly in the paper, maybe by saying, that the policy is trained first by constructing a training set xyz, and at test time it is deployed to edit the prompts. Referencing directly Algorithm 1 would also help a lot.
	- In line with the previous comment, it is hard for me to understand whether K in the training set corresponds to the number of examples for in-context demonstrations, or an external training set, that contains description examples and query. When I think about a few-shot dataset for NLP I am thinking about in-context learning, but here it seems otherwise?
	- Figure 2 would benefit a lot from a walkthrough example. It is hard to understand what exactly is a verbalizer, E1, ... En etc.
	- Section 3.2 could be condensed, a lot of it is already in related work.
	- Clarify, in sec. 4, "... contains both information", what is both information?
	- How is the initial instruction provided? Is it task specific?
	- A figure of the main methods from prompt-search vs the existing method would be helpful to quickly grasp the main differences, or maybe a table of features.
	- Minor: It would be useful in 3.2 to repeat the difference between He et al. 2022, Deng et al. 2022 for section 3.2 since authors talk about query-dependent prompts and then point at limitations of query-agnostic prompts. This seems addressed in the next paragraph, but it is difficult to follow form there that previous methods are these 2 methods. In fact, from my understanding He at al. 2022 is prompt specific, but Deng et al. 2022 is not. Could authors confirm and correct citation in 3.2 in that case?
	- How is Tempera No-TTE different than promptRL. One main thing is the parametrization of the policy, is there something else?
- Method:
	- I would appreciate more details of the policy, there is attention but how is it done? Is it flat over all the techniques and candidates, or is it hierarchical in that first a technique is selected and then a candidate.
	- Table 4 would be good to include No verb, but Inst
- Results
	- Regarding test-time efficiency in Sec. 5.1, while the claim could be true, it is not validated that the run-time will be smaller. A table with clock time or num ops would be good to validate, and if it cannot be provided, make clear that it is a hypothesis.
	- Few-shot is a very valid setting, but it would be really interesting to see if the current method also improves zero-shot performance.

**Summary Of The Paper:**

This work explores editing prompts in large language models on a per-instance level via reinforcement learning. In contrast to existing works, the editing is conditioned on the input query, rather than a task, and the action space for editing is hand-designed. The approach is tested on different natural language few shot classification tasks, showing better performance than few-shot fine-tuning and discrete prompt optimization methods, being more data-efficient for tuning, interpretable, and robust to different hyperparameters in prompting.

**Summary Of The Review:**

This paper discussed a relevant topic, has strong ideas, a new method, thorough experiments, and SOTA results. I generally think it would be a valuable contribution to the community, but in the current state there are important details that are difficult to understand for me, and I assume for other readers in the community. If the aspects described above are clarified, I am very open to changing my rating to accept.

[Edit] The authors have addressed my concerns in the revised version. I therefore have updated my score.

---

> ### Author Response · Authors · 2022-11-16
> **Author Response**
>
> We thank the reviewer for their positive review and thoughtful feedback. The reviewer is mainly concerned with writing clarity and the reproducibility of the paper.
>
> **Q: Does the model tend to balance the number of exemplars per class, or does it try to select more exemplars close to the query class?**
>
> **A:** We thank the reviewer for this very interesting question. According to our initial observation, the selection can be close to the query class. Similar observations can also be found in [1], where the k nearest neighbor can improve the performance of in-context learning. We think it is essential to study rigorously whether there exists a general pattern of example selection. Due to the short period of the rebuttal, we haven't got time to conduct the experiment. We plan to investigate this and update the paper in our next revision.
>
> **Q: Clarify this more explicitly in the paper about the difference between test-time editing and test-time optimization.**
>
> **A:** Thanks for the suggestion and we will add more clarification on the test-time policy. We agree with the reviewer that our method doesn’t do any test time optimization over the policy or prompt. But we conduct test-time editing through a learned RL policy, making the prompt dependent on the query, so we called our method test-time prompt editing. We clarified this point in the paper in the Section. 3.2.
>
> **Q: Whether K in the training set corresponds to the number of examples for in-context demonstrations.**
>
> **A:** We clarify our setting here. We always use an additional fixed set (16 examples in total randomly sampled from the entire training dataset) to choose the in-context demonstrations, and the set is fixed for both training and testing. When training the policy, we use a standard few-shot dataset (16 examples per class), which is different from the fixed set stated above. Note that we never use the fixed set to choose demonstrations for any training. We have clarified this in Section. 5, the task setting paragraph.
>
> **Q: Figure 2 would benefit a lot from a walkthrough example.**
>
> **A:** We thank the reviewer for the information. We have already added an example under “Verbalizer 1”. We have updated the figure in the paper to make it more clear. We also include the instructions, verbalizers and examples in Table. 10 and Table. 11 in Appendix. D.
>
> **Q: Section 3.2 could be condensed.**
>
> **A:** We thank the reviewer for the feedback. We just want to highlight the concept of test-time editing. We have updated the paper accordingly.
>
> **Q: Clarify, in sec. 4, "... contains both information", what is both information?**
>
> **A:** We are sorry for the confusion. It should be “... contains text information”. We have updated it in the paper.
>
> **Q: How is the initial instruction provided? Is it task-specific?**
>
> **A:** We use the initial instruction from the Natural Instructions. It is task-specific[2].
>
> **Q: A figure of the main methods from prompt-search vs the existing method would be helpful.**
>
> **A:** We thank the reviewer for the suggestion. We have included a figure comparing different methods in Figure. 5, Appendix. F.
>
> **Q: It would be useful in 3.2 to repeat the difference between He et al. 2022, Deng et al. 2022.**
>
> **A:** We thank the reviewer for the suggestion. He et al. 2022 is a query-specific prompt, and Deng et al. 2022 is not.  We have updated the paper accordingly in Section. 3.2.
>
> **Q: How is Tempera No-TTE different than promptRL?**
>
> **A:** We summarize the difference between No-TTE tempera compared to promptRL:
>
> 1. No-TTE TEMPERA still performs editing given the original prompt, while promptRL directly generates the tokens.
>
> 2. Given the first point, No-TTE tempera still follows the format of instruction + few-shot examples, while the format of promptRL can be very arbitrary.
>
> 3. The parameterization of the policy is different.
>
> **Q: I would appreciate more details of the policy.**
>
> **A:** We thank the reviewer for pointing this out. Our policy is just flat attention over possible techniques and candidates, and putting a 2-layer MLP on top of the representation. We have updated the paper in the Appendix. B for these details.
>
> **Q: Table 4 would be good to include No verb, but Inst.**
>
> **A:** In Table 4, we mainly ablated the 3 major categories (related to instruction and in-context examples, respectively) of editing tools to see their relative effectiveness (e.g., what’s the performance gain of adding or removing a specific method). We agree it will be more convincing to include more ablations. It is just due to the time interest, we didn’t include all of them. We’ll try to incorporate more ablations in our next revision.

---

> ### Author Response · Authors · 2022-11-16
> **Author Response**
>
> **Q: Regarding test-time efficiency in Sec. 5.1, while the claim could be true, it is not validated that the run-time will be smaller.**
>
> **A:** We thank the reviewer for pointing this out. But since black-box-optimization and grips select a query-independent prompt at training time, there might be no baselines to compare a test-time search method. One simple baseline is performing these searches for each independent query but this might be quite expensive (it might take a couple minutes to achieve the task and we don’t have a ground truth label). In contrast, our policy only runs 10 inferences (which takes less than 1 second) at test time to edit the prompt. We have revised our claim to make it more concrete in Section. 5.1.
>
> **Q: Few-shot is a very valid setting, but it would be really interesting to see if the current method also improves zero-shot performance.**
>
> **A:** We thank the reviewer for pointing out the zero-shot setting. However, our policy needs some data to train it so it might not be possible to evaluate our method in a zero-shot setting. We conjecture the zero-shot performance is very similar to using instruction from natural instructions, randomly selecting 4 in-context demonstrations and a random verbalizer.
>
> However, we add another experiment with the scaling results (512 samples per class). Due to the time limit, we choose 512 samples rather than the full dataset. Results show that our method, in general, outperforms finetuning of the entire model. We would like to highlight this result since it shows, on average especially, RL-based discrete prompt editing can surpass finetuning the entire model with the same amount of data. We have updated the paper on this result in Appendix C, Table. 12.
>
> |          | sst2         | mr       | ag_news   | rte       |
> |----------|--------------|----------|-----------|-----------|
> | Finetune (512-shot per class) | 0.934       | 0.871  | **0.895**     | 0.679    |
> | TEMPERA  (512-shot per class) | **0.938** | **0.887** | 0.888 | **0.7142** |
> |          |              |          |           |           |
> **Q: Clarifying aspects of the method would help in reproducibility, as well as providing the code.**
>
> **A:** We thank the reviewer for asking for the code. We uploaded a zip file containing our code.
>
> We hope our response answer all the concerns raised by the reviewer. We are happy to answer further questions from the reviewer.
>
> **Reference**
>
> [1] Liu, Jiachang, et al. "What Makes Good In-Context Examples for GPT-$3 $?." arXiv preprint arXiv:2101.06804 (2021).
>
> [2] Mishra, Swaroop, et al. "Cross-task generalization via natural language crowdsourcing instructions." arXiv preprint arXiv:2104.08773 (2021).

---

> > ### Comment · Reviewer_Hk2t · 2022-11-28
> > **On Few-Shot**
> >
> > When I mentioned zero-shot I was talking about a case where there are no examples in the initial prompt, and only language descriptions. In theory, the Instruction and Verbalizer ops could still be applied in this case. What would prevent from applying this baseline?

---

> > > ### Author Response · Authors · 2022-11-28
> > > **Author Response**
> > >
> > > We are sorry to misunderstand your point. This is the ablation we plan to add for Tempera without examples. We’ll manage to run some experiments and post the results. We’ll revise the paper in our next revision.

---

> ### Author Response · Authors · 2022-11-18
> **Looking forward to your feedback**
>
> Dear Reviewer,
>
> We hope that you've had a chance to read our response. We would really appreciate a reply as to whether our response and clarifications have addressed the issues raised in the review, or whether there is anything else we can address.

---

### Official Review · Reviewer_Hybx · 2022-10-25

**Confidence:** 4
**Correctness:** 2
**Technical Novelty And Significance:** 3
**Empirical Novelty And Significance:** 3
**Recommendation:** 8

**Clarity, Quality, Novelty And Reproducibility:**

The paper is clear and has some technical novelty. I recommend the authors release the code.

**Strength And Weaknesses:**

## Strengths
1. The paper is overall well-written and easy to follow.
2. The paper includes a wide range of baselines, both continuous and discrete.

## Weaknesses
1. The scalability of the proposed method is not good. As shown in Figure 3, TEMPERA holds an advantage against fine-tuning but cannot scale. Also, I would like to see other few-shot prompting methods illustrated in the same figure.
1. Figure 1 is misleading. It is the best-performing dataset in Figure 3. It's inappropriate to single it out and put it in the introduction.
1. The editing patterns are not diverse enough. The case study is not very convincing. The improvement doesn't seem explainable.
1. The proposed method is in essence similar to prompt/verbalizer searching.
1. How do you justify the use of RL here? Is there a strong reason not to use search instead?
1. I think GLUE or SuperGLUE can be more convincing. Yelp, MR, CR and AG News are considered easy.

-----
After rebuttal: The authors have addressed many concerns mentioned in my review during the response period. I've updated my score to 8.

**Summary Of The Paper:**

This paper proposes RL-based test-time editing for

**Summary Of The Review:**

This paper proposes test-time prompt editing with RL but may have some issues.

---

> ### Author Response · Authors · 2022-11-16
> **Author Response**
>
> We thank the reviewer for their positive review and thoughtful feedback. The reviewer is mainly concerned with the solidness of the experiment, the use of RL in our method, and the potential misleading of Figure. 1.
>
> **Q: The scalability of the proposed method is not good.**
>
> **A:** We thank the reviewer for bringing up the absolute accuracy. We would like to argue that the few-shot performance is an important setting by itself. Prior works also mainly focus on this setting ([1, 2, 3]). In order to test the stability of our method, in addition to the 16-shot per class performance, we add another experiment with the scaling results (512 samples per class). Due to the time limit, we choose 512 samples rather than the full dataset. Results show that our method, in general, outperforms finetuning of the entire model. We would like to highlight this result since it shows, on average especially, RL-based discrete prompt editing can surpass finetuning the entire model with the same amount of data. We have updated the paper on this result in Appendix C, Table. 12.
>
> |          | sst2         | mr       | ag_news   | rte       |
> |----------|--------------|----------|-----------|-----------|
> | Finetune (512-shot per class) | 0.934       | 0.871  | **0.895**     | 0.679    |
> | TEMPERA  (512-shot per class) | **0.938** | **0.887** | 0.888 | **0.7142** |
> |          |              |          |           |           |
>
> **Q: Other methods in Figure. 3.**
>
> **A:** We thank the reviewer for pointing out. We have updated Figure. 3.
>
> **Q: Figure 1 is misleading.**
>
> **A:** We thank the reviewers for the question and would like to clarify the details of the compared method here. We have updated Figure 1 with both 32 examples and 512 examples, averaging over 4 different datasets. We use SST2 as an example in the original Figure. 1 to show how much more samples fine-tuning needs to achieve the same performance as our method, but we realize it is not fair since it is only 1 task. We have updated the figure with an average number of all 4 datasets (SST2, AG News, RTE, and MR), and the general trend still holds (our method is much more sample-efficient compared to fine-tuning to reach the same performance)
>
> **Q: The editing patterns are not diverse enough. The case study is not very convincing. The improvement doesn't seem explainable**
>
> **A: ** We would like to kindly remind the reviewer that our policy provides query-dependent prompts. So for different queries, the prompt can be entirely different. By combining different editing tools in our action space, we can perform instruction editing in phrase level (suppose there are M total phrases in the instruction, there are M! different combinations. M here usually is 6-7.), in-context example selection (similarly, if there are N examples and we want to select 4 examples from them, there are $N(N-1)(N-2)(N-3)$ different combinations. We choose N to be 16 in our experiments). When we calculate the entire space, there are over $3\times10^8$ different combinations for each query.
>
> We include the natural instructions and verbalizers in the Appendix. D. We think that in addition to the in-context example selection, the phrase-level instruction editing combined with the choice of verbalizer can be very diverse.
>
> **Q: The proposed method is similar to prompt/verbalizer search. How do you justify the use of RL here? Is there a strong reason not to use search instead?**
>
> **A:** We would like to kindly remind the reviewer that we perform query-dependent prompt editing. It is not possible to perform such query-dependent editing via prompt/verbalizer search. Because it is not only very time/compute consuming, but also at test time, we don’t have the ground truth label to perform the search. From this perspective, since it is not feasible to use prompt/verbalizer search, it is natural to learn a RL policy to perform test-time query-dependent editing. Please also refer to Figure. 2, Algorithm. 1 and Section. 3.2 for the details.

---

> ### Author Response · Authors · 2022-11-16
> **Author Response**
>
> **Q: I think GLUE or SuperGLUE can be more convincing.**
>
> **A:** Thanks for the suggestion. We have also provided some GLUE and SuperGLUE results in Appendix C. We also would like to kindly note that the RLPrompt paper[1] only compares five datasets: SST2, Yelp, MR, CR, and AG News in their paper.
>
> |                   |                       |     RTE     |    QNLI    |     SNLI     |    MNLI    |    MRPC    |
> |:------------------|:----------------------|:-----------:|:----------:|:------------:|:----------:|:----------:|
> | Finetuning        | Finetuning (few-shot) | 58.6 (3.9)  | 60.2 (4.7) | 54.64 (9.7)  | 47.8 (7.5) | 77.4 (3.6) |
> | Continuous Prompt | Soft Prompt Tuning    | 54.7 (10.9) | 49.7 (0.2) | 36.13 (14.6) | 33.2 (0.0) | 51.6 (0.9) |
> |                   | Black-Box Tuning      | 52.6 (0.9)  | 48.8 (0.6) | 46.58 (1.3)  | 42.9 (2.0) | 61.6 (0.9) |
> |                   | Manual Prompt         |    51.6     |    50.8    |    31.11     |    51.7    |    67.4    |
> |                   | In-Context Demo.      | 60.4 (0.7)  | 53.8 (0.4) | 47.11 (1.4)  | 53.4 (1.5) | 45.8 (0.8) |
> | Discrete Prompt   | Tempera (ours)        | 60.3 (2.2)  | 57.4 (1.5) |  56.4 (3.2)  | 45.2 (2.0) | 74.0 (1.0) |
>
> We hope our response answer all the concerns raised by the reviewer. We are happy to answer further questions from the reviewer.
>
> **Reference:**
>
> [1] Deng, Mingkai, et al. "RLPrompt: Optimizing Discrete Text Prompts With Reinforcement Learning." arXiv preprint arXiv:2205.12548 (2022).
>
> [2] Sun, Tianxiang, et al. "Black-box tuning for language-model-as-a-service." arXiv preprint arXiv:2201.03514 (2022).
>
> [3] Prasad, Archiki, et al. "Grips: Gradient-free, edit-based instruction search for prompting large language models." arXiv preprint arXiv:2203.07281 (2022).

---

> ### Author Response · Authors · 2022-11-18
> **Looking forward to your feedback**
>
> Dear Reviewer,
>
> We hope that you've had a chance to read our response. We would really appreciate a reply as to whether our response and clarifications have addressed the issues raised in the review, or whether there is anything else we can address.

---

> > ### Comment · Reviewer_Hybx · 2022-11-23
> > **Increasing my score**
> >
> > I have carefully read the new version along with the authors' response. It has addressed many of my concerns and I appreciate the authors for their efforts in making this paper better in such a short time period. I'm increasing my score.
> >
> > By the way, there are some missing references that could be added in the revision:
> > - https://arxiv.org/abs/2204.06305
> > - https://arxiv.org/abs/2209.09401

---

> > > ### Author Response · Authors · 2022-11-23
> > > **Thanks for your feedback**
> > >
> > > We would like to thank the reviewer's positive feedback and for increasing the score. We do really appreciate the reviewer's time and effort in the reviewing process. We'll update the references in our next revision.

---

### Official Review · Reviewer_cYCQ · 2022-10-26

**Confidence:** 4
**Correctness:** 3
**Technical Novelty And Significance:** 3
**Empirical Novelty And Significance:** Not applicable
**Recommendation:** 6

**Clarity, Quality, Novelty And Reproducibility:**

The paper looks good to me since the paper is well written, the proposed model is novel, and the comprehensive results show its effectiveness.

**Strength And Weaknesses:**

**Strengths**

**1. The proposed method is novel and effective.**

The paper proposes a novel method to edit the instance-relevant prompt in the test time via RL. The editing/action space covers a comprehensive set of commonly-used components like instructions, few-shot exemplars, and verbalizers. Results show the effectiveness of the proposed method in most datasets.

**2. The paper is well-written and the experiments are comprehensive.**

**Weaknesses/Feedback**

**The experimental section could be improved.**

Some experimental details are missing. For example, what are the dataset scales for SST-2, Yelp P., MR, CR, and AG News? And what is the value of |Y| in “Task Settings”?

What is the exact number of training examples for the results in Figure 1? What are the details of the standard fine-tuning method?

In Table 4, only three different editing techniques are included. It could be critical to include more editing techniques.

On datasets like SST-2, Yelp P., and MR, the improvements of TEMPERA are not obvious compared to existing baselines.

I have no idea of the details of the compared method in Figure 1. So I am concerned it might not be fair to claim that TEMPERA has greater data efficiency than the standard fine-tuning model. Look forward to clarifications in the rebuttal phase.


**Summary Of The Paper:**

The paper proposes a new model that is able to construct the instance-dependent prompt through reinforcement learning. It allows the agent to perform different editing techniques to update instructions, few-shot exemplars, and verbalizers at test time to construct query-dependent prompts efficiently. Results on different datasets show the effectiveness of the proposed model.

**Summary Of The Review:**

The paper is competitive and I am looking forward to seeing my questions answered in the rebuttal stage and in the revised paper.

---

> ### Author Response · Authors · 2022-11-16
> **Author Response**
>
> We thank the reviewer for their positive review and thoughtful feedback. The reviewer is mainly concerned with the clarity of the Figure. 1 and the performance gain of TEMPERA is small.
>
> **Q: Some experimental details are missing.**
>
> **A:** We thank the reviewer for the question and will add more clarification on this. We have updated the paper and also pasted the answer below for clarification. Please see Appendix. E for the details.
>
> **Q: What is the exact number of training examples for the results in Figure 1? What are the details of the standard fine-tuning method?**
>
> **A:** The exact number in Figure. 1 is 32, 64, 128, 256, and 512 samples per class and SST2 has two class labels. We use standard finetuning of the RoBERTa model from huggingface with 100 epochs, a learning rate of 0.0003, and the optimizer of Adam. We have updated these details in the Appendix. E.
>
> **Q: It could be critical to include more editing techniques in Table. 4.**
>
> **A:** In Table 4, we mainly ablated the 3 major categories (related to instruction and in-context examples respectively) of editing tools to see their relative effectiveness (e.g., what’s the performance gain of adding or removing a specific method). We agree it will be more convincing to include more ablations, but we would like to kindly argue that these editing methods already cover a large possible prompt space. Due to time constraints, we didn’t include all of them. We’ll try to incorporate more ablations in our next revision.
>
> **Q: On datasets like SST-2, Yelp P., and MR, the improvements of TEMPERA are not obvious compared to existing baselines.**
>
> **A:** We agree that on datasets like SST2, Yelp and MR, the improvement is smaller compared to AG News and CR. That’s partially because these datasets are relatively easy (as some of the baselines and our method can already reach about 90% accuracy). We would like to note that even a 1.8% improvement in SST2 and 1.3% in MR is quite significant since RLPrompt only improves the previous baseline Black-Box-Optimization by 1.0% and 0.1%, respectively. Our performance gain is also averaged across 4 random seeds and is statistically significant. In addition to that, on average of these 5 datasets, our method improves the previous SoTA RLPrompt by 2.8%.
>
> **Q: I have no idea of the details of the compared method in Figure 1.**
>
> **A:** We thank the reviewers for the question and would like to clarify the details of the compared method here. We have updated Figure 1 with both 32 examples and 512 examples, averaging over 4 different datasets. We use SST2 as an example in the original Figure. 1 to show how much more samples fine-tuning needs to achieve the same performance as our method, but we realize it is not fair since it is only 1 task. We have updated the figure with an average number of all 4 datasets (SST2, AG News, RTE, and MR), and the general trend still holds (our method is much more sample-efficient compared to fine-tuning to reach the same performance). The new results are listed in the table below, and we have updated the paper of this result in Appendix C, Table. 12.
>
> |          | sst2         | mr       | ag_news   | rte       |
> |----------|--------------|----------|-----------|-----------|
> | Finetune (512-shot per class) | 0.934       | 0.871  | **0.895**     | 0.679    |
> | TEMPERA  (512-shot per class) | **0.938** | **0.887** | 0.888 | **0.7142** |
> |          |              |          |           |           |
>
> We hope our response answer all the concerns raised by the reviewer. We are happy to answer further questions from the reviewer.

---

> ### Author Response · Authors · 2022-11-18
> **Looking forward to your feedback**
>
> Dear Reviewer,
>
> We hope that you've had a chance to read our response. We would really appreciate a reply as to whether our response and clarifications have addressed the issues raised in the review, or whether there is anything else we can address.

---

### Official Review · Reviewer_F772 · 2022-11-04

**Confidence:** 3
**Correctness:** 4
**Technical Novelty And Significance:** 3
**Empirical Novelty And Significance:** Not applicable
**Recommendation:** 6

**Clarity, Quality, Novelty And Reproducibility:**

* This paper is clear, novel and seems to provide a lot of implementation details (in appendices).


**Strength And Weaknesses:**

Strength:
* This paper proposes a novel method: It formulates discrete prompt optimization as using RL to edit an initial prompt. It also proposes carefully designed action space, and a set of techniques to improve the final performance.
Weakness:
* It seems that this method mostly improves sample efficiency but not absolute performance (e.g. not in few-shot setting).


**Summary Of The Paper:**

* This paper proposes a test-time prompt editing technique with reinforcement learning. Compare to prior methods, TEMPERA can efficiently leverage prior knowledge, adaptive to different queries, and provides an interpretable prompt for every query. This method achieves 5.33x on average improvement in sample efficiency when compared to traditional fine-tuning methods.


**Summary Of The Review:**

This paper proposes to treat prompt tuning as a test-time editing problem with RL, which can give more flexibility and demonstrate promising empirical results. However, it's novel because it's the first to apply RL on prompt editing, but the RL technique itself is not much novel and seems like it mostly improves sample inefficiency but not absolute accuracy (e.g. not in few-shot setting). So, I'd recommend weak accept.

---

> ### Author Response · Authors · 2022-11-16
> **Author Response**
>
> We thank the reviewer for their positive review and thoughtful feedback. The reviewer is mainly concerned with the stability of our method and its novelty.
>
> **Q: The method is novel because it's the first to apply RL on prompt editing, but the RL technique itself is not much novel.**
>
> **A:** We would like to kindly argue that instead of applying RL for finetuning the entire model, this paper proposes to use RL to select tools for editing the prompt at test time. Compared to prior work (e.g., RLPrompt), our method can easily incorporate human prior (e.g., providing the initial prompt and designing the editing pattern) and is more efficient in editing (rather than generating) the prompt. Although the RL techniques are not novel, we believe this setting of applying for selecting editing tools at test time for an individual query is a novel contribution. This helps connect the RL and large-language model research. In addition, the design of reward shaping and action space of RL is also another contribution of the paper.
>
> **Q: The method seems like mostly improves sample inefficiency but not absolute accuracy.**
>
> **A:** We thank the reviewer for bringing up the absolute accuracy. We would like to argue that the few-shot performance is an important setting by itself. Prior works also mainly focus on this setting ([1, 2, 3]). In order to test the stability of our method, in addition to the 16-shot per class performance, we add another experiment with the scaling results (512 samples per class). Due to the time limit, we choose 512 samples rather than the full dataset. Results show that our method, in general, outperforms finetuning of the entire model. We would like to highlight this result since it shows, on average especially, RL-based **discrete** prompt editing can surpass finetuning the entire model with the same amount of data. We have updated the paper on this result in Appendix C, Table. 12.
>
> |          | sst2         | mr       | ag_news   | rte       |
> |----------|--------------|----------|-----------|-----------|
> | Finetune (512-shot per class) | 0.934       | 0.871  | **0.895**     | 0.679    |
> | TEMPERA  (512-shot per class) | **0.938** | **0.887** | 0.888 | **0.7142** |
> |          |              |          |           |           |
>
> We hope our response answer all the concerns raised by the reviewer. We are happy to answer further questions from the reviewer.
>
> **Reference**
>
> [1] Deng, Mingkai, et al. "RLPrompt: Optimizing Discrete Text Prompts With Reinforcement Learning." arXiv preprint arXiv:2205.12548 (2022).
>
> [2] Sun, Tianxiang, et al. "Black-box tuning for language-model-as-a-service." arXiv preprint arXiv:2201.03514 (2022).
>
> [3] Prasad, Archiki, et al. "Grips: Gradient-free, edit-based instruction search for prompting large language models." arXiv preprint arXiv:2203.07281 (2022).

---

> ### Author Response · Authors · 2022-11-18
> **Looking forward to your feedback**
>
> Dear Reviewer,
>
> We hope that you've had a chance to read our response. We would really appreciate a reply as to whether our response and clarifications have addressed the issues raised in the review, or whether there is anything else we can address.

---

### Decision · Program_Chairs · 2023-01-20

**Decision:**

Accept: notable-top-25%

**Justification For Why Not Higher Score:**

Reviewers felt the experimental setup could have been stronger

**Justification For Why Not Lower Score:**

Strong, novel solution for an important and interesting topic.

**Metareview: Summary, Strengths And Weaknesses:**

The paper proposes a method for learning (using reinforcement learning) to construct instance-specific discrete prompts in real-time for better task performance. Using their method, an agent learns to perform different editing techniques to update instructions, few-shot samples, and task verbalizers to construct query-dependent prompts.

**Strengths:**
Reviewers noted that the method is novel, important, and empirically effective, demonstrating improvements on a diverse set of classification benchmarks.

**Weaknesses:**
Reviewer complaints centered on two points: writing quality, which the authors have put in effort to fix as part of the rebuttal period, and the conclusions about data efficiency of TEMPERA, which the authors seem to have answered to the reviewer’s satisfaction.

Given the universal recognition of the strengths of the paper and the mitigation of outlined weaknesses during the rebuttal period, I think this paper should be accepted.


**Note From Pc:**

if the above contains the word "oral" or "spotlight" please see: "oral" presentation means -> notable-top-5% and "spotlight" means -> notable-top-25%. As stated in our emails, we are disassociating presentation type from AC recommendations

**Summary Of Ac-Reviewer Meeting:**

N/A